# Recent Progress on Synthesis, Characterization, and Performance of Energetic Cocrystals: A Review

**DOI:** 10.3390/molecules27154775

**Published:** 2022-07-26

**Authors:** Manzoor Sultan, Junying Wu, Ihtisham Ul Haq, Muhammad Imran, Lijun Yang, JiaoJiao Wu, Jianying Lu, Lang Chen

**Affiliations:** 1State Key Laboratory of Explosion Science and Technology, Beijing Institute of Technology, Beijing 100081, China; mansultan_gr8@yahoo.com (M.S.); 3120160157@bit.edu.cn (L.Y.); w15313862180@163.com (J.W.); fhxyljy@sohu.com (J.L.); chenlang@bit.edu.cn (L.C.); 2Department of Physics, The University of Lahore, Lahore 54000, Pakistan; muhammad.imranphys2@gmail.com; 3School of Chemistry and Chemical Engineering, Beijing Institute of Technology, Beijing 100081, China; ahtesham.rana@yahoo.com; 4Beijing Key Laboratory of Environmental Science and Engineering, School of Materials Science and Engineering, Beijing Institute of Technology, Beijing 100081, China

**Keywords:** energetic materials, cocrystallization, detonation performance, characterizations of ECCs

## Abstract

In the niche area of energetic materials, a balance between energy and safety is extremely important. To address this “energy–safety contradiction”, energetic cocrystals have been introduced. The investigation of the synthesis methods, characteristics, and efficacy of energetic cocrystals is of the utmost importance for optimizing their design and development. This review covers (i) various synthesis methods for energetic cocrystals; (ii) discusses their characteristics such as structural properties, detonation performance, sensitivity analysis, thermal properties, and morphology mapping, along with other properties such as oxygen balance, solubility, and fluorescence; and (iii) performance with respect to energy contents (detonation velocity and pressure) and sensitivity. This is followed by concluding remarks together with future perspectives.

## 1. Introduction

Energetic materials (EMs), generally composed of carbon, hydrogen, oxygen, and nitrogen atoms, are a class of substances that release a huge amount of energy in a transient manner while undergoing a self-decay phenomenon [1,2]. Classified as explosives, propellants, and pyrotechnics, EMs find a wide range of applications both in civil and military sectors [3,4]. Trinitrotoluene (TNT), 1,3,5,7-tetranitro-1,3,5,7-tetrazacyclooctane (HMX), and 2,4,6,8,10,12-hexanitro-2,4,6,8,10,12-hexaazaisowurtzitane (CL-20) are some of the classic examples of EMs [5,6]. The efficacy of EMs is evaluated based on their characteristics, such as detonation performance, sensitivity, thermal stability, density, and oxygen balance. Due to their molecular structure and the nature of their building units, EMs are generally very sensitive towards shock impact, friction impact, thermal impact, and electric impact. To alleviate the magnitude of these impacts and for effective utilization of EMs, an appropriate balance between energy density and sensitivity of EMs must therefore be established. For that purpose, various strategies have been attempted, including, but not limited to, the enhancement of the crystals’ quality, the invention of new Ems, the doping of EMs by inert materials, and the surface modification of energetic particles by encapsulation with an appropriate coating material [3,7].

EMs produce an abundant amount of energy on detonation but also possess high sensitivity, especially to impact. The imbalance between the characteristic high energy density and augmented impact sensitivity of EMs triggers safety concerns, referred to in the buzzwords “energy–safety contradiction,” needs to be addressed to guarantee safe operations during the receipt of useful energy. The synthesis of new EMs is cumbersome due to complex reaction steps and escalated processing time. On the other hand, crystal modification via coating and doping may reduce the sensitivity of EMs but at the cost of higher energy losses and limited applicability [3,8]. As an alternative strategy to produce materials possessing high energy and lower sensitivity, or in other words, to allow for the energy–safety contradiction, cocrystallization is introduced as a promising technique to produce enhanced EMs, generally referred to as energetic cocrystals (ECCs) [9]. Some of the energetic materials are presented in Figure 1.

Cocrystallization refers to the orderly modification of the molecular structure of two or more elemental crystals without deteriorating the original bonding structure of the constituent crystals. The orderly arrangement of heterogeneous molecules having a fixed stoichiometry is the quintessence of cocrystallization. The resultant product thus formed is termed a ‘cocrystal’ and if one of the constituent crystals is an energetic material, the consequent cocrystalized product is called an ‘energetic cocrystal’ (ECC). ECCs are synthesized in an attempt to bridge the energy–safety contradiction of EMs, and have this as one of their core objectives [3]. For synthesis of an ECC, it is not necessary to synthesize a new energetic material; an existing EM can also be cocrystalized to form an ECC for appropriate applications. Cocrystallization, therefore, is a useful method for tuning the properties of Ems and producing designer, target-oriented EMs [10,11].

Extensive work on research frontiers has been undertaken in recent years to explore different combinations of cocrystals and the main focus has remained on synthesis techniques, the investigation of characteristics, and, to some extent, applications of the developed cocrystals. A major breakthrough in the development of ECCs is the formation of a CL-20/TNT ECC [12,13] wherein both the energy density and impact sensitivity of the resultant ECC is mediocre as compared to the respective characteristics of individual constituents. Another significant study on ECC reports the formation of a CL-20/HMX cocrystal whose energy output is higher and sensitivity is comparable with HMX in its individual capacity [14,15]. Thus, this CL-20/HMX cocrystal is superior from an energy perspective while leaving the safety factor unaffected [16]. Similarly, another cocrystal based on HMX and nitroguanidine (NQ) has been prepared using a vacuum freeze-drying operation to achieve a cocrystal with better thermal stability [17].

Another study reports the utilization of three EMs and inert coformers to produce ECCs with much improved impact sensitivity and thermal stability [17,18]. The coformers can either be energetic or non-energetic in nature. Major coformers used for cocrystallization of EMs include, but are not restricted to, nitramine [19], nitrated azole [20], and nitrobenzene [21]. The cocrystallization of EMs, for example, CL-20, with energetic coformers yields energetic–energetic cocrystals whereas the cocrystallization of CL-20 with non-energetic coformers such as an MMI cocrystal [22] or 2,4-MDNI cocrystal [23] yields energetic–non-energetic cocrystals. Some of the ECCs exhibit noteworthy characteristics either in terms of energy density or impact sensitivity. One such example involves the formation of CL-20/H_2_O_2_ ECC, which possesses a huge amount of energy as compared to the energy of its constituent components individually [24]. By contrast, DADP/TITNB ECC outperforms with respect to impact sensitivity when compared with the individual constituents [25,26]. However, such extreme characteristics are accompanied with augmented sensitivity or an alleviated energy in ECCs, which are highly undesired [27,28].

Millions of research dollars are being spent on the investigation and development of new ECCs across the globe every year. Nevertheless, the research circles of this particular niche area face manifold challenges, such as those related to (i) stability and sensitivity, (ii) safe and economical production [29], (iii) the risk of the emergence of new unwanted structures during the course of ECC development, (iv) the interdependence of ECCs’ compositions with their corresponding structures and performance, (v) little-known growth mechanisms of ECCs via crystal nucleation, (vi) the lack of reliability and validation of the theoretical basis for ECC production due to the formation mechanisms being unknown, and (vii) a vague mechanism of energy release that can potentially limit the industrial applications of ECCs [3].

Recently, several review articles have been published that discuss various aspects of EMs and ECCs. Bennion and Matzger [29] discuss the cocrystallization of TNT, HMX, and other materials with energetic and non-energetic coformers, with special emphasis on the intermolecular interactions, sensitivity of resultant ECCs towards external stimuli, and impact of oxygen-rich and oxygen-deficient cocrystallizing molecules on detonation performance. Another review encompasses thermodynamic aspects, molecular stacking, and intermolecular interactions such as halogen and hydrogen bonding, and π interaction between the molecular cocrystals of energetic materials [14,30]. Different methods for the preparation of ECCs, specially based on CL-20 and HMX, their formation mechanisms, and their thermodynamic, detonating, and structural properties are also discussed in another review by Xue et al. [3].

This review paper covers the philosophical underpinnings of cocrystallization with special emphasis on synthesis, characterizations, and the different properties of the most recent energetic cocrystallizing materials reported in 2020 and 2021. The subsequent sections of this review encompass synthesis techniques for new energetic cocrystals, characteristics of ECCs, thermodynamic aspects of energetic cocrystallization, and conclusions and future perspectives.

## 2. New Energetic Cocrystals

### 2.1. Synthesis Techniques

Cocrystal preparation processes include solid-state grinding, solution-reaction crystallization, solvent evaporation, and slurry conversion, and have all been extensively reported to date. Numerous synthesis techniques have been utilized for the creation of pharmaceutical cocrystals [31,32,33,34,35,36,37,38]. In addition, pharmaceutical cocrystals are easier to produce than ECCs. Therefore, several ECC synthesis methods have been adapted from pharmaceutical cocrystal synthesis methods. The choice of an appropriate cocrystallization technique must still be made empirically. The two types of cocrystal-formation techniques that are most frequently employed are known as solution-based methods and solid-based methods. High solvent consumption is necessary in solution-based procedures in order to dissolve the cocrystal components. Additionally, the choice of solvent has an impact on the cocrystallization outcomes, since it might alter the interactions between EMs and the coformer molecules. In contrast, solid-state techniques have the ability to reduce or even eliminate the need for solvent in the cocrystal-synthesis process. These cocrystal-synthesis methods are discussed briefly here and also given in Figure 2.

#### 2.1.1. Solvent Evaporation

The basic principle of the solvent-evaporation technique for the synthesis of ECCs is that the constituent entities taking part in cocrystallization must have solubilities close to each other. If the participating materials have substantially different solubilities, the component with a lower solubility is likely to precipitate much faster than the other component, thereby leading to the formation of a mixture of solid cocrystal and other components [39]. In some cases, this can result in a complete collapse of the cocrystallization phenomenon. Therefore, care must be taken while choosing the cocrystallizing components to produce ECCs via the solvent-evaporation technique. In this method, the component cocrystals or coformers are dissolved in a solvent as per a pre-defined stoichiometric ratio with subsequent evaporation of the solvent in a sluggish manner to receive the final energetic cocrystals [40]. This method for ECC production is efficacious and cost-effective; however, these advantages are achieved at the cost of a few disadvantages. For instance, (i) this is not an environmentally friendly technique, because if the solvents used are toxic, they may impart hazardous vapors into the atmosphere if a proper disposal system is not installed; (ii) it takes a longer time to process, because the synthesis has to be accomplished at a lower rate of solvent evaporation; (iii) it is accompanied with augmented energy consumption because the evaporation is carried out at an escalated temperature; and (iv) it is difficult, at times, to accomplish the evaporation step in a controlled manner [41]. One of the examples of ECCs formed by the solvent-evaporation technique is the cocrystallization of HMX/AP, which is accomplished by the slow evaporation of the mixture. The hygroscopicity of AP and oxygen balance of HMX are simultaneously improved as a result of using this procedure. 

The solvent-evaporation method is sometimes integrated with vacuum freeze-drying or spray-drying facilities to enhance the safety and quality factors. In vacuum freeze-drying, the solvent is removed from the solution of cocrystallizing components by freezing the solution with subsequent sublimation via the application of vacuum. The remaining solid is the cocrystallized energetic material. This method is used generally for heat-sensitive materials. Vacuum freeze-drying is relatively simple and safe because the volatility of hazardous vapors is much lower and the concentration of solvent is not observed during the course of the crystals’ precipitation [42]. However, this is an expensive technique due to the intensive energy consumption of the freezing process and its longer processing time. The product cost, therefore, increases manyfold, restricting its scaling and commercialization.

Another method to achieve ECCs is by solvent evaporation in a spray dryer, generally equipped with a two-fluid pneumatic nozzle, that introduces the suspension in the form of a very fine spray that comes across a co-current or countercurrent of hot air for the rapid removal of the solvent with the subsequent receipt of the final product from the bottom of a spray dryer [43]. The processing time in this case is much shorter than vacuum freeze-drying and other methods. The product achieved is a very fine powder with a narrower size distribution. The procedure is simple because the additional steps of product purification and solid–liquid separation are not required. However, spray drying is not environmentally friendly or safe because of the generation of some static electricity during the spray-drying process. The encapsulation of HMX by rather insensitive TATB (2,4,6-triamino-1,3,5-trinitrobenzene) particles is one such example of the coating of energetic materials via solvent evaporation in a spray dryer [44].

#### 2.1.2. Solvent/Nonsolvent

Solvent/non-solvent is a frequently used method for the production of ECCs due to its convenience, simplicity, and safety. In this method, the precursor solution is prepared by the agitation of the cocrystallizing components in a solvent. This is followed by the introduction of a nonsolvent that gets the job done either by crystallization or coating of particles to precipitate the cocrystals. Despite its simplicity and convenience, the large quantity of the solvent used poses serious concerns with respect to the quality control of the finished product [43].

#### 2.1.3. Cooling Crystallization

This is a quite simple and environmentally friendly method of cocrystallization. In this method, the crystallizing components, which must have a higher solubility, are dissolved in a solvent and the solution is cooled to an extent to achieve the state of oversaturation [45]. From this point onwards, the solute components cocrystallize and undergo a growth mechanism. In this method, at times, the solvent is one of the cocrystallizing components; for example, the cocrystal of pyridine and quinol is formed when quinol is dissolved in a predetermined volume of pyridine and cooled to form the desired cocrystal [46].

#### 2.1.4. Grinding Methods

In this method, the energetic cocrystals are synthesized by mixing the components in a proportionate manner followed by processing in a ball mill or a mortar to receive the final cocrystallized product. No solvent is involved in the dry-grinding method; however, wet grinding involves the addition of a minimal quantity of solvent for cocrystallization. The dry-grinding method is suitable only for the production of small quantities of ECCs. The addition of a solvent in wet grinding facilitates the enhancement of the reaction rate, crystallinity, and efficiency of the production of the final cocrystals. Moreover, a solvent-assisted grinding method is better for energetic cocrystal formation, as it reduces the friction and heating during the synthesis of the cocrystals, which can be dangerous due to the sensitivity impact of energetic materials. Solvent-assisted grinding is an environmentally friendly method; however, it is difficult to control the cocrystals’ morphologies [43]. L. Yan et al. reported an energetic cocrystal (HNIW/TNT) synthesis using a solvent-assisted grinding method [47]. In their work, they used ethanol as a solvent due to its environmentally friendly nature.

#### 2.1.5. Melting/Condensation Crystallization

The constituent components of cocrystallization are mixed as per the stoichiometric ratio and cooled below their melting temperature to form the cocrystals. Occasionally, the components evaporate and condense back to get the ECCs. This method is suitable for explosives with a broad difference in their melting and decomposition temperatures, such as TNT. It is not suitable for components that have higher melting and lower decomposition temperatures because such materials can undergo thermal decomposition. This method is highly efficient and, therefore, can be used for the industrial production of cocrystals. It is environment friendly because organic solvents are not utilized in this method [43].

#### 2.1.6. Resonant Acoustic Method

This technology harnesses resonance to establish a highly efficient mixing operation for the cocrystallizing components. Since no baffles, impellers, propellers, or other moving parts are involved, this technology provides contactless mixing at the cost of lower energy consumption. In addition to meeting the functional requirements of the ECCs, this technique also caters for the safety requirements since it is highly unlikely to encounter a dangerous stimulation when cocrystallization is accomplished using a resonant acoustic method. With a stoichiometric ratio of 2:1, the production of CL-20/HMX energetic cocrystals is a typical example of this technique. This method has the added advantages of augmenting the mixing efficiency and enhancing the uniformity of the finished product [48].

#### 2.1.7. Slurry Method

This method is comparatively simple in its operation. The constituent components, in a predetermined proportion, are gently stirred in a minute quantity of the solvent that acts as a mediator for the cocrystallization to take place. The slurry is continuously stirred until the reaction is completed and the ECC is formed. In this method, the solvent selection is critical, whereas the solubility factor is not critical [49].

#### 2.1.8. Solvent-Suspension Method

CL-20/HMX cocrystallization with a stoichiometric ratio of 2:1 has been employed to produce ECCs successfully by using the solvent-suspension method, which is simple, safe, environmentally friendly, and produces cocrystals of higher crystallinity and a narrower particle-size distribution. In this method, a solvent (nothing other than deionized water) is used for cocrystallization. The components are added in deionized water and stirred for a long time (generally several hours) at a specific temperature with the subsequent filtration of the desired ECCs. This method is viable for the scaled-up production of ECCs [50].

#### 2.1.9. Self-Assembly Method

This is an innovative method recently introduced for solvent-induced self-assembly of (i) a single energetic crystal with a non-energetic coformer, and (ii) both energetic components for the synthesis of ECCs. This technique involves (i) crystal particles’ induction, (ii) the aggregation of particles in an organized orientation, (iii) the integration of particles’ surfaces, and (iv) the formation of the ECC. Since the impact of heat and mass transfer operations is only minor in this method, it has an inherent convenience in the scaled-up production of ECCs. For the purpose of reference, Figure 3 describes the whole process of the self-assembly synthesis protocol of ECCs. Moreover, the details of all the synthesis methods are depicted in Table 1.

### 2.2. Characteristics of ECCs

In this section, we discuss the most significant characteristics of ECCs, including structural properties, detonation performance, sensitivity analysis, thermal properties, morphology mapping, and other properties such as oxygen balance, solubility, and fluorescence. The investigation of these characteristics paves the way for the efficacy analysis of the existing ECCs and design of new ECCs. In almost all of the studies reported in 2020–2021, the structural properties are investigated by Fourier transform infrared spectroscopy (FTIR), X-ray diffraction (XRD), Raman spectroscopy, or Hirshfeld surface analysis. The detonation performance of ECCs is analyzed by the determination of detonation velocity, detonation pressure, or density concentration of the energetic molecules. One of the most important characteristics of ECCs is the sensitivity analysis of ECCs against external stimuli, be they impact sensitivity, friction sensitivity, spark sensitivity, or electric-field sensitivity. The sensitivity analysis is accomplished by the Bruceton method, BAM fall hammer method, and others. Thermogravimetric analysis (TGA) and differential scanning calorimetry (DSC) are mostly employed for the analysis of thermal stability and decomposition temperatures, while scanning electron microscopy (SEM) is generally used for the morphological study of the ECCs. All these characteristics, in addition to solubility, oxygen balance, and fluorescence, are discussed here.

#### 2.2.1. Structural Properties of ECCs

The structural properties of ECCs display the philosophical underpinning of the molecular or atomic orientation of the cocrystals, which helps to determine the sensitivity and density concentration, thereby facilitating the design of new ECCs. There are different kinds of interactions that take place between the cocrystals, such as CH···O hydrogen-bonding interactions, CH···N hydrogen-bonding interactions, and NO_2_−π interactions. For reference, these three types of interactions are shown in Figure 4. CL-20 is a well-known energetic material that is cocrystallized with several other coformers or energetic materials to synthesize ECCs or energetic–energetic cocrystals (EECCs). For example, ɛ-CL-20 is cocrystallized with TNT and the structural properties of the resultant ɛ-CL-20/TNT cocrystal are investigated using Hirshfeld surface analysis and reduced density gradient (RDG) analysis [52,53]. The benzene ring of TNT becomes an electron deficient π-system due to the strong polarizing effect of the nitro groups. A nitro group of CL-20, therefore, locates itself just above the center of the TNT benzene ring that holds the crystal structure intact via p-π stacking. The ɛ-CL-20/TNT ECC formation is driven by the O-H and N-O interactions while the ECC is stabilized by the O-O interactions. The Raman spectra, densities, and simulated lattice parameters in this study are in synchrony with the experimental values [54,55].

The structural analysis of nano-CL-20/TNT (synthesized in a mechanical ball mill) reveals that ball milling does not alter the molecular orientation of the constituent materials. The resulting nano-CL-20/TNT features a novel crystal phase that differs from the crystal phase obtained by simple mixing [5]. The Hirshfeld surface analysis of 1:2 CL-20/benzaldehyde ECC indicates that overwhelmingly weak hydrogen bonding is the major driving force for the formation of CL-20/benzaldehyde ECC and stabilization of the crystal structure. The main molecular interactions in the crystal lattice include the O—H, O—N, and O—O interactions forming 60.2%, 15.3%, and 17.3%, respectively, of the surface area in the cocrystal. The cocrystal structure also witnesses additional O—C interactions between constituent components of the ECC [57]. Single crystal XRD and powder XRD of two energetic–energetic cocrystals (EECCs) with 1:1 and 1:3 CL-20/1-methyl-4,5-dinitroimidazole (4,5-MDNI) show strong intermolecular affinity between CL-20 and 4,5-MDNI in the form of hydrogen and NO_2_-π bonding that stabilizes the cocrystal structure. Different stacking orientations of the CL-20 and 4,5-MDNI also facilitate the stability of both the EECCs [58]. The crystal structure of ECCs can also be investigated using an evolutionary algorithm (USPEX) coupled with forcefields or ab initio calculations [59]. Similarly, the Hirshfeld surface analysis of CL-20/benzaldehyde ECC reveals the formation of cocrystals by strong hydrogen bonding with a triclinic system [57,60]. For the purpose of reference, the intermolecular interactions of CL-20 with TNT, BTF, and HMX are shown in Figure 5.

Octahydro-1,3,5,7-tetranitro-1,3,5,7-tetrazocine (HMX), the most powerful military explosive, has been cocrystallized in several new studies to make ECCs. For example, HMX has been cocrystallized with N,N-Bis(trinitroethyl)nitramine (BTNEN) to form HMX/BTNEN ECC. The cocrystallization of HMX and BTNEN changes the electron density due to the hydrogen bonding of the resulting ECC. The crystal structure of HMX/BTNEN is different than the structures of individual coformers such that the new positions of ECC diffraction angles are 7, 13, 14.2, 19.7, and 33.2 degrees [62]. Density functional theory (DFT) is employed to study the crystal structure of HMX/FOX-7 (1,1-diamino-2,2-dinitroethylene) ECC. The hydrogen bonding in this case strengthens the N-NO_2_ bonding while increasing the bond-dissociation energy of N-NO_2_ [63]. Another HMX-based ECC was prepared by the solvent/non-solvent method with an insensitive explosive, 6-diamino-3,5-dinitropyridine-1-oxide (ANPyO). The XRD spectrum of the simple mechanical mixture of components appears to be merely a superposition of the individual components, whereas the XRD spectrum of the ECC of HMX/ANPyO is entirely different, thereby implying the formation of the cocrystal. A strong hydrogen bonding exists between the ―NH_2_ of ANPyO and ―NO_2_ of HMX in the HMX/ANPyO ECC. The HMX molecule is replaced by an ANPyO molecule into the crystal lattice [64].

Benzotrifuroxin (BTF) is cocrystallized with various coformers such as trinitrobenzene (TNB), TNT, trinitroaniline (TNA), trinitrobenzene methylamine (MATNB), and 1,3,3-trinitroazetidine (TNAZ). The powder XRD and single XRD analysis reveals that the ECC formation is mainly governed by strong hydrogen bonding [65] in addition to p—π and π—π stacking interactions. The six-membered ring of BTF consists of an electron-poor π-system, so it is natural that BTF would have higher chances of producing ECC with compounds that have a higher number of electron-rich groups [10]. Rapid cocrystallization by the use of differential solubility is employed to synthesize two ECCs, namely, TNB/2,4-MDNI and CL-20/1-methyl-3,4,5-trinitropyrazole (MTNP), and the analysis of intermolecular interactions reveals that both of these ECCs possess stronger intermolecular interactions that are governed by the nitro—π bonding [66]. Figure 6 shows the intermolecular interactions between BTF and different coformers. 

Furthermore, 4,4,5,5-tetranitro-2,2-biimidazole (TNBI) is cocrystallized with fifteen coformers and the characteristics of four of them are thoroughly investigated. The structural analysis reveals that the cocrystal formation is driven by the hydrogen bonding of N—H…N and N—H…O between TNBI and the corresponding coformers. The crystallographic investigation further suggests that an optimum oxygen balance driven by N-oxide-based acceptors produces much better energetic materials. The shelf life and stability of these ECCs is also improved due to their imperviousness to humidity and, therefore, these ECCs can substitute for TNBI materials in industrial applications [67]. The absence of N—H protons in the resulting ECC lowers the hygroscopicity and chemical acidity of the parent compound thereby enhancing its handling, storage, and transport [68].

Maximizing the intermolecular interactions by any means, such as hydrogen bonding or π-stacking, can be utilized for the synergistic detonation performance of EMs. With this intention, a 1:2 ECC is produced from 4H, 8H-difurazano[3,4-b:3′,4′-e] pyrazine and hydroxyl-amine coformers. The resulting ECC displays characteristics akin to 1,3,5-triamino-2,4,6-trinitrobenzene especially the detonation properties are found to be much better than the mechanical mixture of the constituent components. In addition, the detonation performance of the ECC appears to be superior to theoretical prediction, providing it with synergistic properties. This is achieved by (i) appropriate pairing of the cocrystallizing molecules, (ii) developing setups that have H-donor and -acceptor sites, (iii) electron deficient and rich π-systems that ultimately result in an increase in the density of the consequent ECC through strong intermolecular interactions [69].

An ECC based on a 6:1 cyclopentazolate anion (NH_4_N_5_) and ammonium chloride is synthesized by employing the slow-solvent-evaporation method. The crystal structure indicates cube-shaped NH_4_N_5_/NH_4_Cl cocrystals that are formed by hydrogen bonding. The cocrystals are formed mainly by the N—H···N and N—H···Cl hydrogen bonds, and π-π interactions. Due to this crystal structure, the ECC is found to have a higher decomposition temperature, lower sensitivity, and improved detonation performance as compared to the individual coformers [70]. In addition to this, there several cocrystals that have been reported in 2021 and 2022 which are made up of CL-20, BTF, and HMX. For reference, we have given only their names in Table 2.

#### 2.2.2. Detonation Performance of ECCs

The efficacy of an EM or ECC is determined by its detonation performance, which is measured in terms of various factors such as detonation velocity, detonation pressure, and crystal density. Various methods are reported in the literature for the evaluation of the detonation performance of ECCs. Rothstein and Petersen propose a simple, empirical proportion between detonation velocity (*D*) at theoretical maximum density and detonation factor (*F*) that is only based on chemical composition and structure for perfect C, H, N, O-type explosives [88,89]. The detonation factor *F* is given as
(1)$$D=\frac{F-0.26}{0.55}$$

A comparable crystal density of CL-20 and TNT results in an augmented density of the CL-20/TNT cocrystal [54]. The detonation pressure and detonation velocity of a CL-20/benzaldehyde ECC is found to decrease as compared to the pristine CL-20. However, the impact sensitivity of this ECC is decreased. Therefore, such an ECC is suitable for applications where a lower impact sensitivity is required despite a poor detonation performance. The detonation velocity in this case (7455 m/s) is lower than detonation velocity of CL-20 and TNT [57,90].

Another kind of ECC is the energetic–energetic cocrystal (EECC) in which one EM is cocrystallized with another EM to form an EECC. For example, CL-20/4,5-MDNI ECC is cocrystallized with different ratios (1:3 and 1:1) to study the thermal, morphological, and detonation characteristics of the resulting EECC. The results indicate superior detonation performance for 1:1 EECC as compared to 1:3 EECC. The 1:3 ECC is less sensitive, but its detonation performance (D: 8604 m/s, P: 34.45 GPa), however, is better than the recently introduced insensitive ECC LLM-105 (2,6-diamino-3,5-dinitropyrazine-l-oxide). The impact sensitivity of 1:3 EECC is close to that of LLM-105. The results suggest that the stoichiometric ratio of the EM can be manipulated to design new EECCs with improved characteristics. The 1:3 EECC can be regarded as a new high-energy ECC with low impact sensitivity [58].

FOX-7 is cocrystallized with b-HMX and the resulting ECC density (1.9 g/cm^3^) is found to be a little lower than HMX and higher than that of FOX-7 and the same goes for the detonation performance—that is, the detonation performance of b-HMX/FOX-7 ECC is lower than HMX but higher than FOX. Despite a lower detonation performance (detonation velocity = 9.162 km/s) as compared to HMX, this ECC can still be described as a high-density energetic material and an effective explosive. A comparison of the characteristics of two ECCs, that is, CL-20/MTNP and TNB/2,4-MDNI, indicates that CL-20/MTNP has lower impact sensitivity and an augmented density and detonation velocity as compared to the widely known benchmark HMX, which makes it viable for commercial production [66].

#### 2.2.3. Sensitivity Analysis of ECCs

The sensitivity analysis of EMs is extremely important for the design of explosives. It is imperative to determine the factors and the extent to which they affect the sensitivity of ECCs [91]. An ideal ECC is the one that possesses the highest detonation performance or energy content and the lowest possible sensitivity. CL-20/TNT ECC exhibits lower sensitivity as determined by the radial distribution function (RDF) vibrational analysis. The lower sensitivity is attributed to the p-π stacking of the nitro groups of CL-20 and the benzene rings of TNT that keep the crystal structure intact. An overwhelming polarizing effect of TNT’s nitro groups constitute an electron-deficient π-system with consequent positioning of the ―NO_2_ group of CL-20 exactly above the center of TNT’s benzene ring [54]. Another study on nano-CL-20/TNT ECC reveals reduced friction and impact sensitivities as compared to pristine nano-CL-20 and TNT, thereby suggesting the improved safety and viability of this cocrystal explosive in comparison to CL-20 [5].

The Bruceton method is used to study the impact sensitivity of a 1:2 CL-20/benzaldehyde ECC and the findings imply a reduced impact sensitivity, likely for two reasons: (i) the strong hydrogen bonding produces a stable crystal structure that is immune to sudden shock; and (ii) the layered stacking of the crystal lattice also imparts stability to the CL-20/benzaldehyde cocrystal and, therefore, the friction or shock forces are dissipated by the well-packed layers of the cocrystal, thereby decreasing the impact sensitivity. However, this enhanced impact sensitivity is achieved at the cost of a lower detonation velocity and pressure. Thus, it is suggested that such an ECC can be used for applications where high explosive power is not desired [57].

Sometimes, a catastrophic explosion can take place when the ECCs encounter an external electric field. To avoid this, it is important to understand the impact of electric fields on sensitivity and other properties of ECCs in an external electric field so that preemptive abatement strategies can be designed to combat any unforeseen explosion. Several CL-20-based ECCs including CL-20/BTF, CL-20/DNP (3,4-dinitropyrazole), and CL-20/MDNT (1-methyl-3,5-dinitro-1,2,4-triazole) were subjected to an external electric field to investigate its effect on sensitivity and other characteristics of the ECCs. It was found that CL-20/BTF is the most sensitive ECC because of its augmented chemical reactivity, in addition to having the smallest energy gap due to a positive energy field as shown by the electron structure analysis. The analysis of the bond-dissociation energy (BDE) of N-NO_2_ and H_50_ reveals that an increase in a positive electric field renders the impact sensitivity, smaller BDE, and longer trigger-bond length even more sensitive. The increase in negative nitro group charge in a negative external electric field reduces the sensitivity of the ECCs. The larger the negative electric field, the higher the negative charge of the nitro groups, and the lower the sensitivity [92].

The impact sensitivity of ECCs is a function of the intermolecular interactions and packing density of the molecules. An ECC of 2:1 HMX/BTNEN was subjected to the standard GJB-772-97 method for the evaluation of the impact sensitivity of the energetic cocrystal by drop height resulting in a 50% explosion probability (H_50_). The H_50_ value of the 2:1 HMX/BTNEN ECC is found to be 55 cm—that is, in between the H_50_ values of HMX (63 cm) and BTNEN (50 cm), indicating that the synthesized ECC is less impact-sensitive than pure BTNEN. This decrease is attributed to the enhanced packing density of the ECC molecules due to strong intermolecular interactions. Designer ECCs can, therefore, be produced by manipulating the intermolecular interactions by various means [62]. Another study reports similar results in terms of enhanced hydrogen bonding, resulting in a diminished mechanical sensitivity in the case of a hydrazine 3-nitro-1,2,4-triazol-5-one (HNTO)/ammonium nitrate (AN) ECC [17,93].

BTF is cocrystallized with various coformers such as TNB, TNT, TNA, MATNB, and TNAZ, and the drop-weight impact data reveal that BTF/TNB and BTF/TNT ECCs have much lower sensitivities as compared to pristine BTF. In particular, the BTF/TNB ECC is more significant because it possesses explosive properties comparable with RDX but is less sensitive as compared, which vindicates its viability for explosive applications [10]. FOX-7 is cocrystallized with b-HMX and the density functional theory indicates the existence of hydrogen bonding, which causes strong N-NO_2_ bonding that not only reduces the ECC sensitivity but also increases the bond dissociation energy of N-NO_2_ bonds [63].

#### 2.2.4. Thermal Properties of ECCs

The thermal properties such of ECCs such as their thermal decomposition temperature are important to investigate because they provide useful information about the behavior of a certain ECC with respect to changes in temperature. In addition, it also helps to compare thermal behaviors of pristine components and the ECC. Generally, TGA and DSC are employed to investigate the thermal behaviors of different ECCs. For instance, nano-CL-20/TNT ECC was subjected to DSC, which revealed that the ECC had a higher decomposition temperature, with a broad exothermic decomposition peak at 102–150 °C, as compared to the individual coformers. Hence, the cocrystallization of nano-CL-20 and TNT adds value to the resulting cocrystal in the form of enhanced thermal stability which could not have been possible with the simple mechanical mixing of the components [5]. Similarly, DSC analysis of an EECC (CL-20/4,5-MDNI) with different CL-20:4,5-MDNI stoichiometric ratios indicates improved thermal stability suggesting that the stoichiometric ratios of the ECCs can be regulated to achieve the desired properties of ECCs [58].

A comparative analysis of thermal properties (measured at different pressures) between an ECC of CL-20/HMX and a simple mechanical mixture of CL-20 and HMX reveals that the highest decomposition temperature of the ECC is lower than the individual and pristine CL-20 and HMX despite having a comparable thermal stability when compared with CL-20 only. Nevertheless, the CL-20/HMX ECC exhibits distinct thermal properties as compared to the mechanical mixture of CL-20 and HMX. For example, the heat liberated from the ECC is found to be more concentrated, most likely because of the CL-20/HMX mixture. In addition, the ECC displays a lower combustion rate, attributed to the organized crystal structure characterized by the intermolecular interactions via hydrogen bonding. The pressure exponent of the combustion rate of CL-20/HMX ECC is also lower as compared to individual CL-20 and HMX [72].

Just like other properties, the deflagration (burning rate) properties of ECCs can neither be evaluated by the simple superposition of the deflagration behavior of the individual coformers or the mechanical/physical mixture thereof. For example, the burning rate of a CL-20/TNT ECC and a physical mixture of CL-20 and TNT is found to be in between the burning rate of the coformers, much lower than CL-20. It is, therefore, important to evaluate how cocrystallization affects the burning rate of the cocrystallizing components. In this regard, a study has encompassed the deflagration behavior of 2:1 CL-20/HMX and 1:1 CL-20/TNT energetic cocrystals and their mechanical mixtures as a function of pressure by employing planar laser-induced fluorescence (PLIF) of C—N and O—H interactions. The burning rate of 2:1 CL-20/HMX is found to be close to the burning rate of CL-20 alone, whereas the burning rate of 1:1 CL-20/TNT is in between the individual coformers CL-20 and TNT. The burning rate characterized by the flame structure is found to closely depend on the particle size of the constituent components. For a smaller particle size (<100 µm), the burning rate and flame structure of the cocrystals matches the burning rate and flame structure of physical mixtures of the corresponding coformers. The deflagration behavior of individual coformers does not reflect the deflagration behavior of the mechanical mixture and ECC and the flame structure provides useful information on the burning rate of the components [94].

The kinetics of thermal decomposition provide useful information to evaluate the stability of ECCs. The isoconversional kinetic methods can be employed to study thermal decomposition kinetics via apparent activation energy, reaction mode, and the frequency factor. The amount of energy required by the coformers to initiate the reaction is termed the apparent activation energy. The kinetic investigation of an ECC HNTO/AN synthesized by the solvent-evaporation method and using the isoconversional method for the determination of apparent kinetic energy reveals that the apparent activation energy of HNTO/AN ECC lies in between those of the coformers [93].

#### 2.2.5. Morphological and Allied Properties of ECCs

The surface analysis and particle size and texture on the surface of the ECC particles is also important as the morphological features reveal important information about the sensitivity and detonation performance of the ECCs. In almost all of the studies, scanning electron microscopic micrographs are captured to analyze the energetic materials’ morphology. For example, a CL-20/TNT ECC synthesized in a mechanical ball is subjected to SEM analysis and it is found that particles (~115.9 nm on average) show a sphere-like appearance [5]. The effect of change in the stoichiometric ratios of the ECC while forming an EECC by using 1:3 and 1:1 CL-20/4,5-MDNI cocrystals indicate that the density of the packing layers of final EECC can be regulated by altering the stoichiometric ratios of the ECCs. The same applies for the morphology of the EECC based on CL-20/4,5-MDNI [58].

Since HEMs are very sensitive, even traces of these materials can be significantly dangerous. Therefore, it is necessary to develop techniques that can be used to detect even traces of ECCs. One such technique that has gained attention recently includes the use of fluorescence sensing for ECC trace detection. For example, functionalized fluorophore polyaniline is employed for the trace detection of an ECC based on CL-20 and RDX using electrochemical and spectroscopic techniques. The results indicate that the limit of detection (LOD) value of the CL-20/RDX ECC is very low as compared to other high-energy materials, implying that this technique is useful for the trace detection of ECCs. Since the nitro groups of ECCs are embedded in a cage-like structure, the access of the functionalized fluorophore to these nitro groups is limited; therefore, these nitro groups play little role in quenching the fluorescence emission of fluorophore. Therefore, the functionalized fluorophore is effective for trace detection of ECC only at a lower concentrations, for which the nitro groups are high in number [95].

The solubility and temperature dependance of the solubility of HEMs is another important characteristic that needs to be investigated, because many ECCs are synthesized in solvents. A study has reported the solubility and its temperature dependence for CL-20 and HMX in 29 solvents and some of their mixtures at 293.15 K and 333.15 K. In most of these cases, the solubility of CL-20 appears to be temperature independent and that of HMX to be strongly temperature dependent. In addition, the solubility of HMX was found to be far lower than CL-20; therefore, CL-20/HMX cocrystallization is proposed only in those solvents that do not form thermodynamically stable solvates of HMX [96]. The scaleup of CL-20/HMX production, probably the most promising ECC, remains a challenge due to the intrinsically augmented solubility difference of the two coformers. Despite being an efficient cocrystallization method, reaction cocrystallization, also known as the slurry technique, has not been systematically exploited for the successfully scaled-up production of CL-20/HMX ECC. Semi-batch reaction cocrystallization (SBRC) has been employed for the production of a 100 g batch of CL-20/HMX with a particle size of 163 µm. The recovery rate of CL-20 in this case is 63%, double that of evaporation crystallization, while the crystal quality is similar to that obtained in a controlled antisolvent crystallization method. The quantity of CL-20/HMX ECC thus obtained in a timely manner can be utilized with an explosive utility that requires an augmented quantity of this ECC. The study indicates the significance of SBRC for the scaled-up production of ECCs [97].

The oxidant-to-fuel ratio is an important characteristic of energetic materials that determines the dynamics of burning and subsequent detonation performance of the energetic materials. One of a series of important studies encompassed the oxygen balance and its impacts on structure and other properties of energetic materials. In this study, moderately strong and oxygen-rich acids (H_5_IO_6_ and HIO_3_) are cocrystallized as oxidants with weak basic energetic materials including 4,4′-bis-1,2,4-triazole (BTRZ), 4,4′-azo-1,2,4-triazole (ATRZ), and 2,4,6-triamino-5-nitropyrimidine-1,3-dioxide (ICM-102) to form four ECCs (H_5_IO_6_/BTRZ, H_5_IO_6_/ATRZ, 2HIO_3_/ATRZ, and HIO_3_/ICM-102) by exploiting the close acid–base gap of the two precursors. The structural characterization of these ECCs shows that all four cocrystals have large oxygen bonds, especially 2HIO_3_/ATRZ, which possesses the largest number of oxygen atoms. This is because oxygen-rich acids (H_5_IO_6_ and HIO_3_) are one of the constituent components of these ECCs. The fourth ECC (HIO_3_/ICM-102) was found to have excellent biocidal capability. The ease of preparation of the ECCs, their promising thermal stability, and, as a cherry on top, their enhanced detonation performance make these ECCs quite attractive for commercial production [98].

The most powerful military explosive, HMX, was cocrystallized by the solvent/non-solvent method with a relatively insensitive energetic material ANPyO to study various characteristics of the resulting ECC. The morphological analysis indicated that this ECC is composed of polyhedron-shaped particles with a density greater than the density of the coformers. The optimum ratios for effective ECC formation are calculated to be 4:1 and 8:1 (HMX:ANPyO). Thermal analysis reveals that HMX/ANPyO ECC decomposes at approximately 285 °C—that is, lower than the decomposition temperature of ANPyO and higher than that of HMX, whereas the enthalpy of formation of the ECC is also higher than that of the individual coformers. The ECC was found to be less sensitive as compared to pristine HMX, which shows its significance [64].

## 3. Thermodynamic Aspects of ECCs

The thermodynamic knowledge of energetic cocrystallization is important for the design and development of new ECCs [3]. For this purpose, it is suggested that the process of ECC development should be thermodynamically spontaneous and thermodynamic parameters such as enthalpy, Gibbs free energy, and internal energy must be taken into consideration []. Another important parameter that provides useful information on the uniformity of ECCs is the change in solubility (Δ*δ*) of the components of an ECC. For example, in pharmaceutics, it is suggested that a lower Δ*δ* is feasible for ECC formation [99]. Similarly, the change in internal or lattice energy (Δ*E*) also provides useful information, such as evidence of ECC formation, that can be vindicated if the resultant ECC is at a lower energy state. Since the crystallization of single components releases less energy as compared to cocrystallization, an ECC at a lower energy state indicates the successful formation of the ECC. The change in enthalpy (Δ*H*) is another thermodynamic parameter that provides information on the miscibility of the coformers participating in the cocrystallization. The miscibility data are important for deciding if the cocrystallization is proceeding in a positive direction or not [61].

The change in Gibbs free energy (Δ*G*) is a thermodynamic parameter that is used to evaluate the relation of thermal stability of the ECC with its pristine coformers. To date, the published literature on ECCs is more about the structural, morphological, thermal, and efficacy analysis of ECCs with respect to detonation performance and has relatively little to do with thermodynamic investigations of ECCs [3]. It is observed that ECC formation is mostly governed by entropy instead of enthalpy. Such systems that are driven by entropy require a choice of an appropriate solvent so as to achieve adequate mixing and subsequent homogeneous ECCs [100]. As an example, the formation of CL-20/1,4-DNI ECC is thermodynamically spontaneous as per the thermodynamic parameters and three-phase diagram analysis. In addition, thermodynamic analysis also reveals that the use of acetone solvent at a low temperature would produce effective results with respect to ECC formation [101].

Ternary phase diagrams provide information about the phase behavior of two pure components dissolved in a solvent in addition to the thermodynamically stable region of all the prevailing phases in the cocrystallization process [102,103]. The choice of a solvent plays a significant role in determining the position and size of a region of a cocrystal phase that is thermodynamically stable [104]. In any solvent/solute system, the symmetry of the phase diagram is a function of the extent of solubility of both pure components for a pure and a mixture solvent system. An asymmetric ternary phase diagram results if there exists a broad difference in the solubility of the pure components in solvent. To the contrary, a symmetric ternary phase diagram is achieved as a result of the consistent dissolution behavior of the cocrystal in the case of the two pure components having solubility close to each other. The ternary phase diagram suggests that the cocrystal can be formed by the adjustment of the starting composition of the solution, which makes the process of cocrystal formation difficult. A symmetric ternary phase diagram, on the other hand, suggests that a cocrystal can be synthesized by cooling crystallization. This implies that the investigation of the symmetry of a phase diagram is important for the manufacture of cocrystals on a larger scale. Both temperature and the solvent have a significant effect on thermodynamically stable region of a cocrystal. The study of the effect of temperature and solvent is, therefore, important [105]. For example, in the case of a 2HNIW/HMX cocrystal synthesis, the ternary phase diagram in ethyl acetate and acetonitrile indicates that the cocrystal has a wide thermodynamically stable region in ethyl acetate but a narrower thermodynamically stable region in acetonitrile, as shown in Figure 7 [106]. In addition, the phase diagram is found to strongly depend on the solvent and has a weak dependence on temperature.

The solubility of pure components in solvent is also an important factor for the design and development of new ECCs. Since temperature affects solubility significantly, it is important to discuss the effect of temperature on the solubility of ECC coformers in different solvents. Generally it is observed that solubility increases with a rise in temperature [3]. This is vindicated in a study that suggests that the solubility of HNIW/TNT cocrystal in nine different solvents (1,2-dichloroethane, toluene, ethanol, butanone, m-xylene, chloroform, methanol, ethyl acetate, and acetonitrile) increases with increases in temperature (between 283.15 K and 318.15 K) such that the solubility of HNIW/TNT ECC in nine solvents is successfully correlated by Yaw’s model, the λh equation, the van’t Hoff equation, and a modified Apelblat equation. These correlations are used for cocrystallization and the thermodynamic investigation of HNIW and TNT coformers [107].

## 4. Conclusions and Future Perspectives

This review article discusses various synthesis techniques, properties, and the performance of energetic cocrystals reported in the literature in 2020 and 2021. Most of these studies have outlined structural properties, detonation performance, sensitivity analysis, thermal properties, and morphology mapping, along with other properties such as oxygen balance, solubility, and fluorescence. All of these properties are important, particularly from the standpoint of the design and development of new energetic cocrystals. Hence, these studies can be considered significant contributions for the extension of research frontiers in the field of energetic cocrystals. However, there are certain issues that need to be discussed to pave the way for more effective research contribution in this niche area. These points are discussed below.

In crystal engineering, this is of utmost importance to establish the relationship between the properties of energetic cocrystals with the composition and molecular structure of the coformers. The future studies need to focus more on this perspective.Many of the current studies are based merely on the grounds of theoretical simulations. However, a real-time investigation and understanding of the actual crystal structure and other properties are direly needed for the design and development of future energetic cocrystals.Studies on the real-time scaleup of the manufacturing of energetic cocrystals are scarce. Rigorous work in this dimension is required to benefit from the fascinating properties of energetic cocrystals.Many of the existing studies are based on trial-and-error strategies when it comes to the optimization of ECC design. Competitive relationships are expected to be developed in future to thoroughly investigate the mechanism of ECC formation for the subsequent design and synthesis of novel ECCs.For ideal energetic cocrystals, it is expected that a synthesized ECC will have a sensitivity less than the sensitivity of the individual coformers and an energy density higher than them. However, only a few studies meet this criterion. In most of the studies, the formed ECC has either higher sensitivity or higher energy density OR lower energy density and lower sensitivity. More work needs to be accomplished to develop new ECCs with characteristics closer to the ideal ones.Several synthesis techniques for ECCs have been described in the literature that were employed many years ago and have never been reported again. Furthermore, several synthesis strategies for pharmaceutical cocrystals have recently been reported. It is advised that similar synthesis approaches be used for ECCs as well.Only few studies have discussed the thermodynamic aspects of ECC production and their relationship with crystal properties and ECC performance with respect to sensitivity and detonation. Researchers in this area are expected to make this particular aspect the central point of their research.

## Figures and Tables

**Figure 1 molecules-27-04775-f001:**
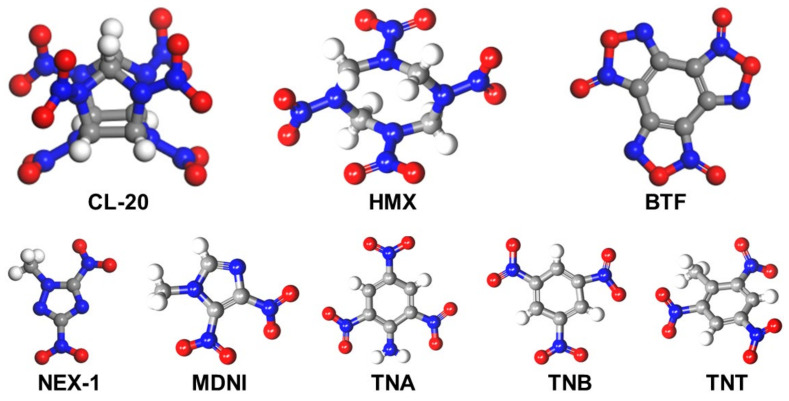
Types of energetic materials and co formers. The carbon, hydrogen, oxygen, and nitrogen atoms are represented in grey, white, red, and blue, respectively.

**Figure 2 molecules-27-04775-f002:**
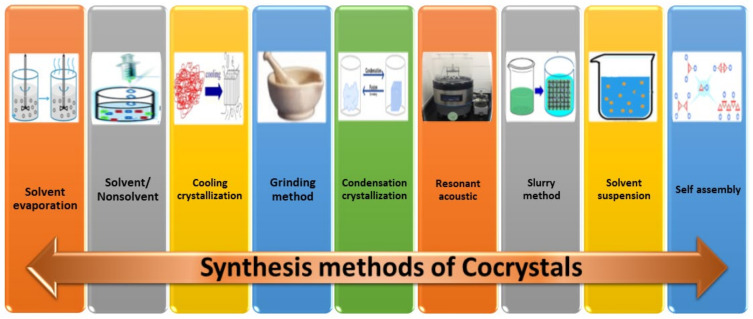
Synthesis methods of energetic cocrystals.

**Figure 3 molecules-27-04775-f003:**
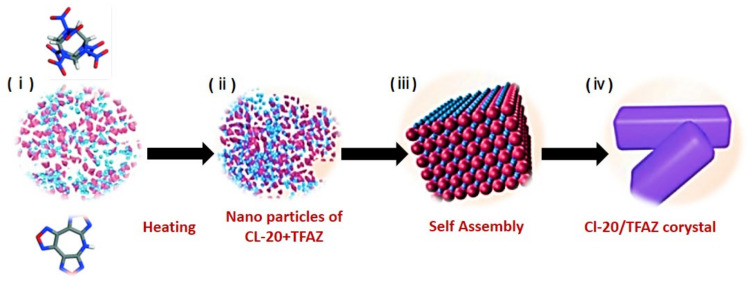
Self-assembly protocol for cocrystal synthesis. Reproduced from [51]. Copyright 2020, The Royal Society of Chemistry.

**Figure 4 molecules-27-04775-f004:**
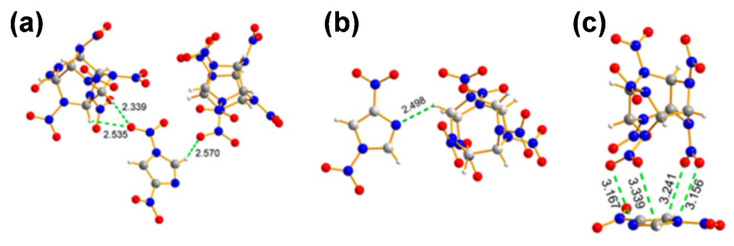
(**a**) CH···O hydrogen-bonding interactions. (**b**) CH···N hydrogen-bonding interactions. (**c**) NO_2_−π interactions. Reproduced from [56]. Copyright 2019, American Chemical Society.

**Figure 5 molecules-27-04775-f005:**
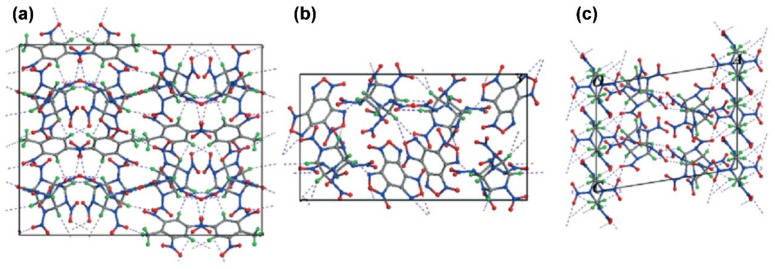
Intermolecular hydrogen bonds in CL-20-based cocrystals (represented by purple dashes). (**a**) CL-20/TNT, (**b**) CL-20/BTF, and (**c**) CL-20/HMX. The carbon, hydrogen, oxygen, and nitrogen atoms are represented in grey, green, red, and blue, respectively. Reproduced from [61]. Copyright 2015, The Royal Society of Chemistry.

**Figure 6 molecules-27-04775-f006:**
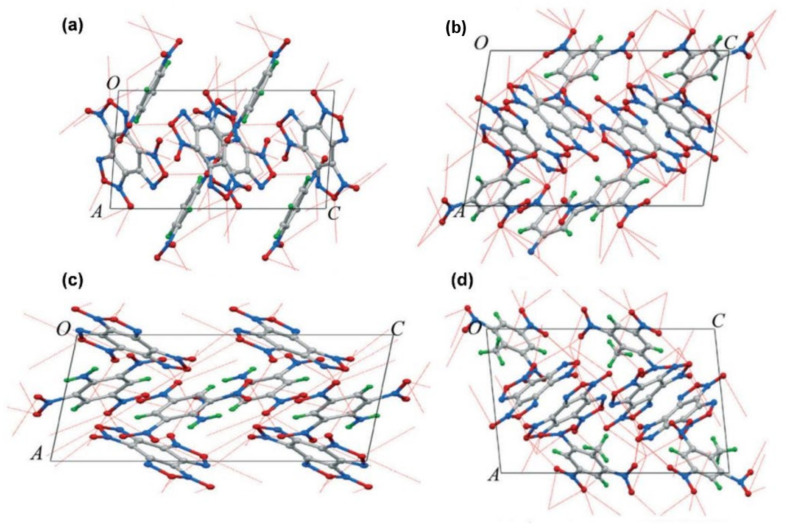
The intermolecular hydrogen bond and N—O⋯NO_2_ type interactions between BTF and DNB, TNB, TNA, and TNB molecules in cocrystals. (**a**) BTF/DNB cocrystal, (**b**) BTF/TNB cocrystal, (**c**) BTF/TNA, and (**d**) BTF/TNT, respectively. The lengths of H⋯N, H⋯O, and O⋯N contacts are presented in red dashed lines. Reproduced from reference [61]. Copyright 2015, The Royal Society of Chemistry).

**Figure 7 molecules-27-04775-f007:**
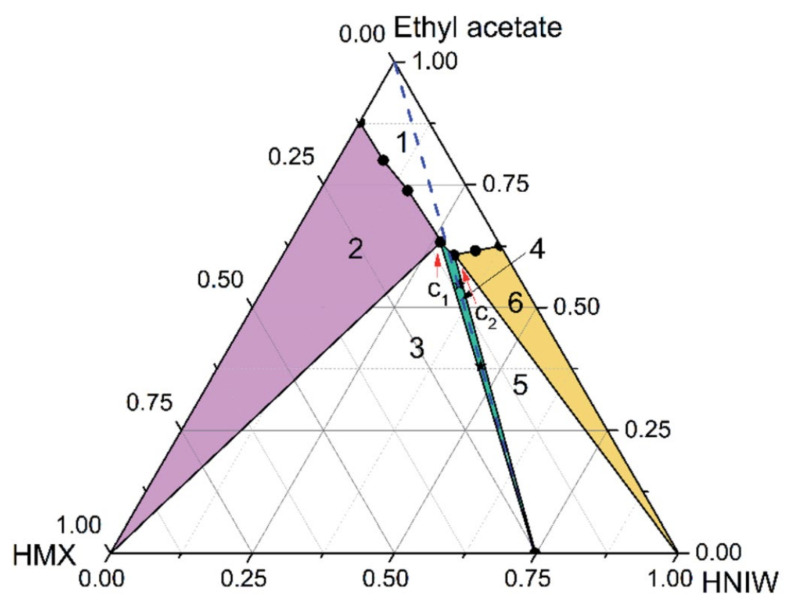
Ternary phase diagram for HMX–HNIW–ethyl acetate system at 15 °C. The points (black stars) represent starting compositions for cocrystals. Reproduced from [106]. Copyright 2015, The Royal Society of Chemistry.

**Table 1 molecules-27-04775-t001:** Synthesis methods of energetic cocrystal formation.

Synthesis Method	Advantages	Disadvantages	Scalability
Solvent evaporation	Efficient, cheap, high purity product, controlled morphology, and safe process.	Requires high temperature, high evaporation time, and is not environment friendly.	Yes, with some modification, e.g., by using the rotary or spray-drying processes.
Solvent/no solvent	Easy, safe, widely recognized and practical.	Excessive solvent used, uncontrolled crystal formation.	Yes
Cooling crystallization	Facile synthesis, environmentally friendly, and widely accepted.	Requires high solubility and raw materials.	Yes
Grinding method	Ecofriendly, less solvent consumption, fast, and consumes less raw materials.	Unsafe, size and morphology are uncontrolled, incomplete cocrystal formation.	No
Condensation crystallization	Fast, efficient, and environmentally friendly.	Decomposition occurs	No
Resonant acoustic	Resource-efficient method; excellent consistency; less hazardous.	High equipment cost and noticeably small manufacturing scale.	No
Slurry method	Easy, less solvent used, safe, and independent of solubility.	Cocrystal quality and controlled morphology are compromised.	Yes
Solvent suspension	Less time-consuming, less harmful to the environment, and high product crystallinity.		No
Self-assembly method	Cost-effectiveness, safe, high yield, and high processability.	Small-scale production	No

**Table 2 molecules-27-04775-t002:** Newly published cocrystal based on CL-20, BTF, and HMX explosives.

Explosive	Co-Former	Published Year	References
CL-20	BTF	2021	[71]
	HMX	2021	[72]
	2,4-DNI	2022	[73]
	TNAD	2022	[74]
	LLM-105	2022	[75]
	Nitroimidazole	2022	[76]
	DNB	2021	[77]
	MTNI	2022	[78]
	N_2_O	2022	[79]
	DNDA5	2021	[80]
BTF	TNAZ	2020	[81]
	TNT	2021	[82]
	TNAZ	2022	[83]
	NB	2022	[84]
HMX	LLM-05	2020	[85]
	ANPyO	2021	[64]
	BTNEN	2021	[62]
	AP	2021	[64]
	NMP	2021	[86]
	DATAD	2022	[87]
	LLM-116	2022	[87]
	Keto-RDX	2022	[87]

## Data Availability

Not applicable.

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
