# Peer review of "Recent Progress on Synthesis, Characterization, and Performance of Energetic Cocrystals: A Review"

_molecules, 2022, doi:10.3390/molecules27154775_

Round 1

Reviewer 1 Report

The review by Wu and co-workers summarizes the progresses done in the area of ECC in the last 2 years. Several aspects are considered from the synthesis, to properties, thermodynamics…

In my opinion, to be published it requires several extensive revisions which are summarized in the following text.

Major revisions

In many parts, the article seems a mere list of things taken from the literature. The purpose of a review should be to make an effort to process data from different works to find trends or indications that help understand how to improve compounds / syntheses / processes / designs ... In this sense, the conclusions help but they just indicate area of developing rather than provide suggestions for developing those area.

More in general, a lot of works as been already done in the pharmaceutical CCs but no mention is done while it would be useful to transfer all this knowledge also in the world of ECC. For instance, a lot of emphasis is given to the synthetic part only because it is applied to EM. However, in the world of CCs applied to drugs and pharmaceutical compounds, these techniques are very well known. In some way it should be stated.

In the paragraph of the Grinding Methods, strangely, there is no mention of the danger of grinding explosive substances if one thinks about their sensitivity to impact, friction and heating all phenomena thhe grinding technique relies on.

In the “Self Assembly Method” paragraph, it is not clear how one produces ECCs; how crystal particles formation is induced, how they organize…

In the paragraph Structural Properties of ECCs, I would reduce the description of the weak interactions or put some figures of the structures that identify the interactions to facilitate the reading that is otherwise very complex.

In the conclusions, concerning the point 4, the authors completely omit all the development of methods for the prediction of co-crystal formation that have been published in the last 3-4 years. See for example the articles by R. de Gelder and co-workers.

In the whole article there are only three figures. Some additional figures would be very useful for better understanding the text: for instance, when describing structure and/or weak interactions or when citing phase diagrams.

Although English in general is good, there are several errors in the text that should be corrected. Review by a native speaker is recommended

Minor revisions:

Lines 58-61: revise the English

Line 141: change “dew” with “few”

Line 149: “AP” was not defined

Lines 292-294: revise the English

Lines 388-389: I do not agree with the definition of EECC “Another kind of ECC is the energetic-energetic cocrystal (EECC) in which one ECC is cocrystallized with another ECC to form an EECC”. In my opinion, an EECC is formed by two EMs rather than by two ECCs. Furthermore, many of the examples cited throughout the text are EECCs (almost all of them; e.g. HMX/BTNEN, HMX/FOX-7…). All these should be defined EECCs.

Line 494: in the sentence “For example, the heat liberated from the ECC is found to be more concentrated that is most likely because of the CL-20/HMX mixture” it is not clear to me what “the heat liberated is more concentrated” means. Could the authors better explain it?

Author Response

Dear Prof. Thank you so much for your valuable suggestions. We have revised our manuscript according to your comments. 

Reviewer 2 Report

I was very curious to read this specific review because the topic is interesting. However, a lot of editing work is still required to turn this manuscript into a publishable review. The paper is not focused and missing valuable information. There are sections that resemble too much other published reviews. The pictures seem insignificant and not related to the topic. Too many examples are given instead of a summary with one or maximum two examples.

In order to turn this manuscript to a real review, you'll need to do changes such as:

1. The manuscript is full of material name Acronyms. It would be nice to have a scheme of all molecules that the manuscript is describing.

2. Figure 1 is copied from another review. You might need the (Acc. in Chem. Res.) journal's consent to include this (same) picture in your review. If this manuscript was mine, I would not use Figure 1 at all. I did not understand figure 3 and there is no reference to it in the text.

3. The synthesis techniques section is almost the same as in the review Cryst. Growth Des. 2020, 20, 12, 8124–8147 https://doi.org/10.1021/acs.cgd.0c01122. The sentences are rephrased, but the examples are the same. This manuscript is about detonating materials.  If this was my manuscript, I would get rid of the examples and add some safety precautions. I would also order the methods according to how easy to perform the crystallisation method, the crystal yield and safety.

4. There are many detailed examples in the structural properties section. It's very hard to follow and understand without pictures. There is no classification or explanation why there is a need to present so many examples

5. There is a formula in the detonation performance section without any explanation how it is derived. It would be nice if you could explain how to measure the F. Where are the values: 0.26 and 0.55 (in the formula) are originated from? 

Author Response

Dear Respected Prof. Thank you so much for your valuable comments, we have revised our manuscript according to your suggestions.

Reviewer 3 Report

This review ("Recent progress on synthesis, characterization and performance 2 of energetic co-crystals: A review") describes recent success in the development of new energetic co-crystals. I think, the present work can be published in journal "Molecules", however, only after some major corrections.

 1. In Section 2.1, authors provide a brief overview on variety of well-known methods for co-crystal obtaining. Evidently, those methods were already described in the literature. At the same time, the present review covers 2020-2021 period, however most references cited in Section 2.1 are pretty old. I strongly recommend to describe those methods which were utilized in articles published during 2020-2021 period, while giving only brief description for others. Also I'm not sure that references on pharmaceutical co-crystals should be mention in the present work. I think it would be better to replace, at least some of them, with appropriate energetic ones.

 2. In Section 2.2.1, authors describe co-crystals obtained for most popular explosives such as CL20, BTF, HMX, etc. I believe that it would be really interesting for Reader to know how many co-crystals were published for each of mentioned popular explosives in 2020-2021 period. I don't think, it is necessary to describe all of them in the details, but just mention. Unfortunately not all of them are in the current version of CSD, but authors can use another services to get necessary info.

 3. There are several misprints in the manuscript. For instance,

P.7, string 278 "known " (not know); P. 15, string 699 "In" (not IN).

I'm not sure that I have caught all of them.

Author Response

Dear Prof., I'm really thankful to you for your valuable suggestions and appreciation. we have changed our manuscript according to your comments.

Round 2

Reviewer 1 Report

The article by Wu and co-workers was extensively changed from the first submission. The authors answered many of the points I raised in my first review. However, in my opinion, there are still some points that have to be improved before publication:

1) Concerning the literature on pharmaceutical cocrystals: I understand that  EMs are the focus of the article, at the same time one cannot totally ignore all the knowledge produced within the same discipline, i.e. crystal engineering, even if acquired on other applications, i.e. pharmaceutical field. I think it is necessary to refer to the pharmaceutical field in some way.

2) Concerning the “Self Assembly Method” paragraph: by looking at the paper cited by the authors (N. Liu et al., “Preparation of CL-20/TFAZ cocrystals under aqueous conditions: balancing high performance and low sensitivity,” CrystEngComm, vol. 21, no. 47, pp. 7271–7279, Dec. 2019, doi: 10.1039/C9CE01221D) I think the authors are misunderstanding what is written. In fact, all the co-crystals obtained through co-crystallizations exploit the "Self Assembly Method". It is not an innovative method at all because it is exactly the method used to obtain most of co-crystals reported in the literature. In this sense, my previous comment on the knowledge gained from pharmaceutical cocrystals is more appropriate than ever. I renew my invitation to take a closer look at the literature on pharmaceutical cocrystals. Furthermore, the paper cites also the seeding method as "Self Assembly Method" for obtaining cocrystals. The general idea is that all methods of synthesis to obtain co-crystals are "Self Assembly Methods".

Author Response

We are very much grateful to you for your time and valuable suggestions. We hope that the current form of the manuscript has improved a lot and may be very fit for the readers of this Journal.

Reviewer 2 Report

I think that this manuscript should be rejected. The authors invested a minimum of time to revise the text. The manuscript is still very hard to read and follow. It contains an endless list of acronyms. Figures are made to help you understand the text better, otherwise you don't need them. There is no correlation between the text to some figures. There is no reference in the text to the other figures that might be relevant. The text is filled with meaningless statements such as in line No. 321, since there is no correlation between a triclinic system and a "strong hydrogen bonding". 

Hydrogen bonds are easy to identify in small molecules. The distances between the hydrogen donor and hydrogen acceptor is smaller than 3.0A and the angle is about 180 degrees. pi-pi stacking is a little more difficult to identify. However, this manuscript is a review and all the structural information appears in the original published structure. It is obvious the authors have no idea and drew random contacts (dashed red lines) between atoms in all pictures.

Author Response

Thank you very much for your time and valuable comments. We believe that the response to your raised comments has improved the quality of this manuscript and made it more fit for this Journal.

Reviewer 3 Report

I'm satisfied with the answers to my first and third questions. At the same time, an answer to the second question is not completed. It is good idea to make a separate Table. However, please check carefully the literature, and you will find some more cocrystals for HMX and BTF published recently which is necessary to include into this Table.

Author Response

We appreciate your time and valuable suggestions to make this manuscript more fit for this Journal. We hope that now this manuscript will provide some scientific information to the readers of this Journal.
